# Expression of Lumican and Osteopontin in Perivascular Areas of the Glioblastoma Peritumoral Niche and Its Value for Prognosis

**DOI:** 10.3390/ijms26010192

**Published:** 2024-12-29

**Authors:** María Dolores Salinas, Pablo Rodriguez, Gonzalo Rubio, Rut Valdor

**Affiliations:** 1Biochemistry, Molecular Biology B and Immunology Department, University of Murcia (UMU), 30120 Murcia, Spain; mdsh@um.es (M.D.S.); grubio@um.es (G.R.); 2Unit of Autophagy, Immune Response and Tolerance in Pathologic Processes, Biomedical Research Institute of Murcia (IMIB), 30120 Murcia, Spain

**Keywords:** glioblastoma, tumor microenvironment, pericyte, perivascular, peritumoral, biomarker, sex differences, CD68, Iba-1, CMA, LAMP-2A, lumican, osteopontin

## Abstract

Glioblastoma (GB) is one of the most aggressive and treatment-resistant cancers due to its complex tumor microenvironment (TME). We previously showed that GB progression is dependent on the aberrant induction of chaperone-mediated autophagy (CMA) in pericytes (PCs), which promotes TME immunosuppression through the PC secretome. The secretion of extracellular matrix (ECM) proteins with anti-tumor (Lumican) and pro-tumoral (Osteopontin, OPN) properties was shown to be dependent on the regulation of GB-induced CMA in PCs. As biomarkers are rarely studied in TME, in this work, we aimed to validate Lumican and OPN as prognostic markers in the perivascular areas of the peritumoral niche of a cohort of GB patients. Previously, we had validated their expression in GB xenografted mice presenting GB infiltration (OPN) or GB elimination (Lumican) dependent on competent or deficient CMA PCs, respectively. Then, patient sample classification by GB infiltration into the peritumoral brain parenchyma was related to GB-induced CMA in microvasculature PCs, analyzing the expression of the lysosomal receptor, LAMP-2A. Our results revealed a correlation between GB-induced CMA activity in peritumoral PCs and GB patients’ outcomes, identifying three degrees of severity. The perivascular expression of both immune activation markers, Iba1 and CD68, was related to CMA-dependent PC immune function and determined as useful for efficient GB prognosis. Lumican expression was identified in perivascular areas of patients with less severe outcome and partially co-localizing with PCs presenting low CMA activity, while OPN was primarily found in perivascular areas of patients with poor outcome and partially co-localizing with PCs presenting high CMA activity. Importantly, we found sex differences in the incidence of middle-aged patients, being significantly higher in men but with worse prognosis in women. Our results confirmed that Lumican and OPN in perivascular areas of the GB peritumoral niche are effective predictive biomarkers for evaluating prognosis and monitoring possible therapeutic immune responses dependent on PCs in tumor progression.

## 1. **Introduction**

Glioblastoma (GB) is the most common primary malignant tumor of the central nervous system (CNS). Molecular biomarkers have recently gained importance in the diagnosis and classification of CNS neoplasms. Therefore, histopathological classification is now complemented by data obtained through molecular genetic studies, providing diagnostic, prognostic, and predictive values. In the case of GB, it is essential to conduct an appropriate molecular study to diagnose and classify the tumor accurately [1]. However, biomarkers are not usually associated with the tumor microenvironment (TME) despite its demonstrated importance in the progression of GB, treatment resistance and tumor recurrence [2,3,4]. Regardless of the high presence of immune cells, TME is characterized by a state of immunosuppression due to the secretion of cytokines by the tumor cells, GB-associated microglia and macrophages (GAMMs) and pericytes (PCs) [5,6,7].

Our team showed that GB cells interact with PCs of peritumoral areas and enhance the chaperone-mediated autophagy (CMA) pathway [8]. This GB-induced CMA in the PC leads to an immunosuppressive phenotype, which allows tumor proliferation and survival [6].

GAMMs are the predominant immune cell types in GB, constituting up to 30–40% of the TME [9,10,11], and are being studied for their role in prognosis prediction [12]. Iba-1 is a pan-mononuclear phagocyte marker, including both activated microglia and macrophages [11]. Importantly, this marker has also been shown in PCs acting as mesenchymal stem cells under hypoxic conditions to give rise to inflammatory microglia [13,14,15]. Additionally, the CD68 marker is usually found to be upregulated in inflammatory myeloid cells like macrophages and activated microglia [9,11]. However, it is also possible to identify it as a PC marker, as PCs express it when they are stimulated with IFNγ and act like macrophages [13,16,17]. We have previously studied these two markers in a GB mouse model in response to treatment with CMA-deficient PCs, showing activation of the anti-tumor innate response in the tumor’s surrounding areas [18]. Therefore, the study of these immune markers in perivascular areas of the GB invasion front might let us understand the TME and relate it with the immune function of PCs during GB progression [8].

On the other hand, extracellular matrix (ECM) and secreted proteins constitute part of the complex array of the TME [19]. Importantly, GB-induced CMA in murine PCs has been shown to increase the secretion of molecules such as Osteopontin (OPN) [18], which supports tumor growth [20,21,22,23], while the inhibition of this CMA upregulation leads to the release of molecules with anti-tumor properties, such as Lumican [18,24,25,26,27,28,29]. Thus, a better understanding of the relationship between the secretion of these two proteins and the GB-induced CMA status in human PCs could let us validate these possible prognostic markers in GB patients.

Lumican is a small proteoglycan of the extracellular matrix with a dual nature depending on the cellular context in cancer pathogenesis [30]. Its pro-tumor [30,31] or anti-tumor effects [18,24,25,27,28,29,30,32] seem to be dependent on its intracellular role or functions when it is extracellularly secreted in the TME [33]. That is the case for Lumican expression by pancreatic cancer cells, which need it to proliferate, while Lumican secreted by stromal cells has been shown to enhance apoptosis of this type of tumor cells [25]. In the context of brain cancers, the relevance of Lumican remains poorly studied. The increased expression of intracellular Lumican in GB and neuroblastoma cells has been linked to the preservation of a stem-like phenotype, drug (temozolomide) resistance, and reduced overall survival [34,35]. In contrast, in medulloblastoma in children, extracellular Lumican is detected in the low-risk group but not in aggressive tumors [36].

On the other hand, OPN is a multifunctional secreted phosphorylated glycoprotein with adhesion sequences to interact with ECM components and cell surface integrins. Through these interactions, OPN regulates chemotactic migration and the survival of macrophages [20,21,22]. In the context of brain tumors, and in GB in particular, OPN secreted by tumor cells and GAMMs contributes to the recruitment of macrophages to the GAMMs pool and to M2 polarization maintenance, which, in turn, compromises the anti-tumor immune responses, favors angiogenesis, and facilitates tumor evasion [21,22,37].

Our previous data suggest that Lumican and OPN could serve as promising biomarkers for prognosis in the progression of GB and that their secretion by PCs is dependent on GB-induced CMA [18]. Therefore, in this work, the extracellular proteins Lumican and OPN were analyzed in the peritumoral and perivascular microenvironments at the GB invasion front of a cohort of patients. The patients were classified according to their prognostic outcome related to the perivascular GB-induced CMA and the immune activation markers Iba-1 and CD68. Our results show a correlation between the perivascular activated phagocytic populations and the patient outcome. We validated the expression of Lumican and OPN as biomarkers of good and poor prognosis (respectively) in the peritumoral niche of the GB invasion front, dependent on the CMA activity in PCs.

## 2. Results

### 2.1. Validation of Lumican and OPN as GB Prognosis Biomarkers in a Mouse Model with GB Infiltration Dependent on GB-Induced CMA in PCs

We have previously shown in a comparative mass spec study that OPN is included in the GB-conditioned PC secretome, which is dependent on aberrantly induced CMA in PCs [8] and is composed mainly of proteins implicated in the pro-tumor immune responses [18]. In contrast, Lumican secretion was found to be enriched in the secretome of CMA-deficient PCs responding to GB cells, among other molecules contributing to the anti-tumor immune responses [18]. Additionally, the RNAseq study of the GB-conditioned PCs compared to CMA-deficient PCs in response to GB cells revealed gene expression profiles that supported previous results on pro-tumoral anti-inflammatory and anti-tumoral inflammatory phenotypes of PCs, respectively. Moreover, our findings revealed that CMA-deficient PCs, through their secretome, can eliminate GB tumor growth, whereas competent CMA PCs promote GB progression [8,18]. Concluding with these studies, we determined that some molecules in the PC secretome, along with the expression of genes related to their phenotype, might be used as possible prognostic markers [18]. Thus, we decided to validate Lumican and OPN in the perivascular areas, including PCs, as GB progression prognostic markers. Firstly, we tested their expression in the GB peritumoral niche of our GB mouse model that presented GB infiltration or GB elimination, depending on competent or deficient CMA PCs, respectively.

As expected, brain areas with GB tumor mass only showed Lumican expression within the tumor cell mass and in the peritumoral GB cell infiltration. In contrast, Lumican was significantly identified in perivascular areas surrounding brain parenchyma where there was a previous tumor mass and infiltrated GB elimination (Figure 1A). Conversely, OPN was found in peritumoral microvessels of GB mice, supporting tumor cell growth and survival through GB-induced CMA in PCs. In contrast, mice presenting GB elimination just showed OPN expression in some remnant tumor cells (Figure 1B). With this first approach, we have proven that Lumican and OPN could be biomarkers of GB progression, dependent on GB-induced CMA in PCs.

To support our results, we aimed to further characterize the link between CMA and Lumican and OPN expression in clinical samples from public datasets (TCGA and Rembrandt cohorts). These datasets had previously shown to present high levels of *LAMP2A* gene expression, which positively correlated with metabolism, proliferation, and ECM interaction markers in glioblastoma stem cells (GSCs) [38].

Importantly, data of GB samples from these two cohorts revealed a positive correlation between *LAMP2* gene expression and both Lumican (*LUM*) (Figure 1C) and OPN (*SPP1*) genes (Figure 1D). Our findings conclusively validated both perivascular Lumican and OPN expression, probably dependent on PC CMA, as biomarkers of the prognosis of GB progression.

### 2.2. Analysis of CMA Activity in Peritumoral PCs of GB Invasion Front Correlates with Patient Overall Survival

A cohort of 34 grade IV GB patients was evaluated (Table 1). Prior to analyzing molecular biomarker expression, an overview of each patient’s tissue characteristics was obtained by reviewing HE and GFAP stained images used for peritumoral area identification. In assessing the level of invasion by GB cells in tissue, we set the borderline between the peritumoral regions and the beginning of the invasion front to establish the areas of GB progression in which to analyze the markers. This histological analysis and the clinical history of the patients were used for patient severity classification, categorizing them as mild, moderate or severe. As we have previously shown that GB-induced CMA in PCs is required for tumor cell survival and tumor progression [8,18], we checked the patient severity classification according to GB-induced CMA in PCs located in peritumoral microvessels, where GB cells infiltrate from the invasion front [8]. Then, we analyzed the expression levels of the CMA lysosomal receptor, LAMP-2A, in peritumoral PCs (PC markers; α-SMA or PDGFRβ) (Figure 2 and Appendix A).

Severe patients displayed hypercellularity with elevated astroglia-invaded parenchyma that correlated with the appearance of aberrant blood vessels (Appendix A). In the peritumoral areas of these patients, PCs showed upregulated levels of CMA activity (Figure 2A,B), as has been previously shown [8]. Similarly, moderate classified patients exhibited large parenchyma invasion, with GB cells becoming round in the peritumoral areas (Appendix AB). In these patients, CMA activity was still high in PCs (Figure 2A,B), but the perivascular areas were not yet fully invaded by tumor cells (Appendix AB). In contrast, mild patients displayed a regular brain parenchyma with few invaded blood vessels (Figure 2A and Appendix A), which was related to an expected normal LAMP-2A expression in PCs (Figure 2A,B). Consequently, the severity grade of patients and, therefore, the GB-induced CMA activity of their peritumoral PCs were correlated with the cohort survival (Figure 2C).

Differences in incidence and survival depending on age and gender in GB have been reported [39], so we analyzed these variables along with severity classification in the present cohort. Most patients in middle age were men, as previously described (Figure 2D), and interestingly, all except one of the female patients were included in the moderate and severe groups. In contrast, the mild group was mostly formed by male patients (Figure 2E).

### 2.3. Perivascular Expression of CD68 and Iba-1 in Peritumoral Areas Is Correlated with Low CMA Activity in PCs and Better Patient Outcome

GAMMs are implicated in GB progression, spreading and angiogenesis [9,10], as well as PCs [6,8,40]. PCs have been found to be associated with microglia [41], with macrophage recruitment [17,42], and to exhibit some characteristics of these two cell types [15]. As we have previously shown an association between GB elimination by CMA-deficient PCs and an activation of inflammatory cells expressing CD68 and Iba [18], we wondered whether the perivascular detection of these phagocytic activation markers [18] would correlate to the immune function of peritumoral PCs with low CMA and, thus, to the mild severity of patients.

Our results showed that the expression of microglia marker Iba-1 was increased in peritumoral areas of patients classified as severe (Figure 3A), with values three times higher than the ones considered mild (Figure 3B). But importantly, the highest average of positive perivascular cells expressing Iba-1 corresponded to mild-to-moderate patients, whereas it was scarcely detected in the perivascular areas of the severe ones (Figure 3A,C). Therefore, there seems to be a negative correlation between the expression of perivascular Iba-1 and the grade of histopathological severity of patients related to CMA activity in peritumoral PCs (Figure 2A,B). In other words, Iba-1 perivascular expression in peritumoral areas is related to low CMA activity in PCs and, thus, to a better outcome in patient survival.

Similarly, infiltrated macrophages and activated microglia expressing CD68 appeared elevated in peritumoral areas in patients classified as severe, appearing to be decreased by up to half in mild-to-moderate patients (Figure 3D,E). Interestingly, perivascular cells of mild-to-moderate cases showed an elevated expression of CD68, losing the expression of this marker in severe cases (Figure 3D,F). Indeed, mild patients only showed positive cells in the perivascular area. These results suggest that a high number of perivascular cells marked with CD68 could also be used as a marker of good prognosis.

Altogether, the expression of both markers Iba-1 and CD68 in the perivascular microenvironment of the peritumoral niche correlates with a favorable prognosis of patients related to low levels of CMA in PCs, revealing a potential use of these markers for monitoring tumor progression (Figure 3G).

### 2.4. Lumican Expression in the Peritumoral Vascular Microenvironment Proves to Be a Good GB Prognostic Marker

The ECM has emerged as a critical part of the TME, influencing resident cells and controlling cancer invasion and metastasis [19,43,44]. The presence of a variety of molecules opens the opportunity to discover new biomarkers for tumor prognosis. In previous studies from our lab, we have shown that murine CMA-deficient PCs have an anti-tumoral secretome in response to GB, enhancing the secretion of Lumican [18]. Thus, we wanted to validate if the expression of this extracellular protein in the perivascular microenvironment of peritumoral areas of GB patients could be related to the absence of GB-induced CMA in PCs and, consequently, could be used as a biomarker.

The histological examination of Lumican expression revealed variations in peritumoral levels related to the patient’s severity according to CMA activity in their PCs (Figure 4A). Mild patients showed lower expression of Lumican in peritumoral areas, which is consistent with a lower infiltration of GB cells in the parenchyma (Figure 4A and Appendix AB). In contrast, moderate and severe patients exhibited twice the expression levels of peritumoral Lumican than mild patients (Figure 4A,B).

Analysis of perivascular Lumican revealed an interesting pattern (Figure 4A): mild patients showed twice the levels of Lumican expression in perivascular areas than moderate and severe patients (Figure 4C). Consistently, mild-to-moderate male patients exhibited higher perivascular levels of Lumican than females (Appendix AA). Importantly, we found partial co-localization of Lumican expression with PCs of mild patients (Figure 4D,E), indicating that perivascular Lumican expression related to deficient levels of GB-induced CMA in peritumoral PCs correlates with a good patient outcome (Figure 2A–C and Figure 4D).

### 2.5. OPN Expression in Peritumoral Perivascular Areas Dependent on PC CMA Proves to Be a Poor GB Prognostic Marker

We have previously described that GB-conditioned PCs through CMA activity promote the secretion of OPN, a molecule included in a pro-tumoral secretome [18]. Furthermore, OPN has been reported as an anti-tumor immune response-compromising chemokine [37]. Thus, we wanted to validate if the expression of OPN in the vascular microenvironment of peritumoral areas of GB patients could correlate with a poor progression outcome related to the GB-induced CMA in PCs.

Analyzing the histological expression of OPN in the peritumoral regions, we observed that mild patients exhibited low positivity (Figure 5A,B). However, moderate and severe patients showed three times higher OPN expression than mild patients (Figure 5B), which is related to the increased infiltration of GB cells. Corroborating previous studies in vitro with murine PCs [18] and related to high CMA activity in peritumoral PCs (Figure 2A,B), we observed that perivascular expression of OPN was four times higher in severe patients than in patients with mild and moderate severity grades (Figure 5A,C). Comparing between genders, the mild-to-moderate severity groups of female patients showing poor outcomes in middle age (Figure 2E) showed higher perivascular levels of OPN than males (Appendix AB). Thus, these results also validate perivascular OPN as an indicator of GB progression with poor prognosis. Moreover, the comparative analysis of the peritumoral expression of Lumican and OPN in the perivascular microenvironment of GB patients corroborated our previous results, as we found a negative correlation in the expression of these markers (Figure 5D). Importantly, we identified that the perivascular OPN expression also co-localized partially with PCs (Figure 5E,F), corroborating previous studies [18] on OPN secretion by GB-conditioned PCs and supporting that perivascular OPN expression is dependent on GB-induced CMA in PCs.

## 3. Discussion

First, we validated the expression of the two ECM proteins, Lumican and OPN, in our GB mouse model (Figure 1A,B). We also correlated their gene expression with the CMA gene *LAMP2* in the TCGA and Rembrandt public datasets (Figure 1C,D).

Our study focused on analyzing the consequences of CMA activity in PCs of the microvasculature surrounding the invasion front of GB and its relationship with the overall survival of patients. For this purpose, a cohort of 34 patients with grade IV GB was analyzed (Table 1). It is crucial to highlight that the evaluation of molecular biomarker expression in patient tissue was performed using a comprehensive approach that combined image review of HE and GFAP staining to identify peritumoral areas (Appendix A) and classify the severity of patients, followed by the co-localized staining of brain microvessels against PC markers (αSMA or PDGFRβ) and the CMA marker LAMP-2A (Figure 2). Corroborating our previous research in a GB mouse model [8], our findings showed that patient classification based on PC CMA activity correlates positively with the degree of peritumoral parenchymal invasion and with poor patient outcome (Figure 2 and Appendix A).

The GB invasion front is dependent on the immunosuppressive function of GB-conditioned PCs through the aberrant induction of CMA activity [6,8]. We do not know if the PCs of the tumor core and the peritumoral zone are different in number, but they might, as we have previously shown [6,8] that GB-conditioned PCs acquire immunosuppressive properties, and their proliferation is affected. We believe that PCs in the tumor core or invasion front seem equal, but their immune functions can present differences depending on the grade of GB invasion. Thus, the immune compartment is determined by the number of GB cells conditioning PCs through aberrantly induced CMA. In the core, all PCs are conditioned by the complete GB invasion of the microvasculature, whereas in the microvasculature of peritumoral regions where GB cells infiltrate, there are still non-conditioned PCs responding to the tumor. Thus, the immune properties of PCs and, consequently, the surrounding tumor cell microenvironment could change between both edge and core compartments [8,45]. Consequently, relating the TME immune status with the degree of GB-induced CMA in peritumoral PCs might be useful in validating possible markers for GB progression and, finally, in improving patient outcomes.

Our results are supported by previous studies [39,45], also showing a higher prevalence in male patients. We have found gender differences in the patient cohort when classified by severity grade (i.e., depending on PC CMA and their consequent PC immune function), specifically from early to late middle age (Figure 2D,E). Interestingly, we found that the female patients presented a worse prognosis, as the cases detected were mainly classified as moderate-severe.

GAMMs are implicated in tumor progression [5,9] and are less studied in peritumoral areas. The study of Iba-1 and CD68 is useful for identifying these myeloid cells and understanding their activation state [11]. In most studies, Iba-1 is proposed as a general marker, but it can also be used to identify cells with greater phagocytic activity and increased motility, which could help to identify activated microglia in the M1 differentiation state [9,11,46]. On the other hand, there are controversial studies for the CD68 marker. Some studies indicate that it could be used to identify M1 macrophages that oppose tumor growth [47], but others suggest that, along with CD163, it can be used as a marker of M2 polarization [48] since its higher expression is associated with decreased survival in GB patients [49].

Our results show that the expression of CD68 and Iba-1 in peritumoral parenchyma areas seems to be related to infiltrated GAMMs in severe cases, as has been described previously in intratumoral areas [46,47]. However, their perivascular expression is related to activated perivascular cells in patients with mild-to-intermediate outcomes. In this work, we show for the first time the association of these markers with peritumoral perivascular cells, such as PCs with low CMA activity, that could be activating their immune function to respond against GB invasion and limit tumor progression before being conditioned by the tumor cells. Therefore, a high expression of Iba-1- and CD68-positive perivascular cells at the microvasculature of the peritumoral niche could be used as an indicator of good prognosis.

As studies from our lab using murine PCs identified two ECM proteins, Lumican and OPN, as possible prognostic markers of GB progression dependent on PC CMA activity [18], we validated their expression in a cohort of patients (Table 1) classified according to their outcome and related to their PC CMA activity. Lumican has been classified as a marker of both good and poor prognoses, dependent on the cancer in which it is expressed [30]. When secreted by stromal cells, Lumican enhanced the apoptosis of pancreatic cancer cells [25]. Although intracellular Lumican has been found to stimulate neuroblastoma cell migration [35], the effect of Lumican secretion by other cells on brain cancers has not been thoroughly evaluated. A recent study in medulloblastoma has described that Lumican can be specifically located on the periphery of nodules of the low-risk subset of medulloblastoma, but it is absent on the nodules of the most aggressive subtype and frequently associated with metastases [36]. Related to this study, anti-tumoral PCs, in response to GB cells, secrete elevated levels of Lumican, as well as other proteins with anti-tumoral properties, which might contribute to reduced GB progression [18].

On the other hand, OPN expression is increasingly being linked to poor prognosis and cancer progression, playing key roles in tumorigenesis and immune evasion [21,22,50]. OPN is found primarily in tumor cells and in tumor-infiltrating myeloid cells, promoting TME immunosuppression [20]. GAMMs and PCs are also a source of OPN [51], which could be favoring the spread of the tumor. Importantly, with our work, its immunosuppressive participation in GB progression is supported by the finding that the perivascular expression of OPN is related to GB-induced CMA in PCs of patients with worse outcomes (Figure 5).

Our findings show that Lumican and OPN are solid indicators of poor prognosis at the peritumoral parenchyma level in GB patients. The expression of both proteins is found at low levels in patients with better outcomes, whereas patients with worse outcomes display elevated expression levels (Figure 4 and Figure 5). Excitingly, and validating previous studies from our lab in the context of the perivascular region of the peritumoral niche, Lumican expression levels are higher in mild and moderate patients than in severe ones (Figure 4), while OPN expression in this area shows a trend of decreased expression (Figure 5). Consistent with previous results on the female patient tendency to present worse outcomes (Figure 2E), in particular in middle age groups, we found that female patients exhibit lower levels of Lumican and higher levels of OPN at the perivascular level (Appendix A). Consequently, these results might add molecular background to the finding of the poorer prognosis of female patients.

Our results are supported not only by the role of Lumican and OPN in polarizing immune cells but also by a possible complementary role in the intravasation and subsequent extravasation of tumor cells in metastasis. OPN is a substrate of extracellular matrix metalloproteinases (MMPs), which are found in increased levels in human GB biopsies compared to normal brains or are expressed by GB cell lines and eventually upregulated by TNF- or TGF-β [52,53,54,55]. Thus, the increased TGF-β secretion by GB-conditioned PCs [6] may be relevant to the MMP increase. The proteolytic activation of OPN by MMPs would enhance its binding to integrins and promote the adhesion and migration of tumor cells, as found in hepatocellular carcinoma [56,57] and non-small cell lung cancer [58]. In the opposite direction is the high expression of perivascular Lumican, inhibiting metastasis by reducing the activation of MMPs important for the activation of OPN [59]. In addition, perivascular Lumican can interfere with integrin signaling, reducing the angiogenic activity of endothelial cells [59] and affecting the blood–brain barrier function [60]. Specifically, Lumican knockdown cells, in this context, showed a reduction in the localization of the protein ZO-1 at cell–cell junctions, indicative of barrier disruption, and showed an increase in permeability. ZO-1 is one of the tight junction proteins that, when its levels decrease, has been related to higher-grade tumors [61,62].

In conclusion, our results validated the expression of Lumican and OPN, two ECM proteins, as possible potential biomarkers of good and poor prognosis, respectively, for perivascular areas of microvasculature surrounding the GB invasion front and depending on PC CMA activity. Lumican and OPN, as biomarkers, could be used to predict the infiltration tendency of the tumor and give information about the possible response of patients to treatments, as these proteins are related to CMA-dependent PC immune function. Importantly, our results reveal that the assessment of CMA in peritumoral PCs of brain blood microvessels serves as a method of patient evaluation. Additionally, as PCs are involved in modulating immune populations, CMA correlates with the expression of perivascular phagocytic markers, which could be used as good indicators for GB prognosis.

### Limitations for the Study

As with any research, ours has several limitations that should be considered when interpreting the results. Heterogeneity of GB: Glioblastoma is known for its heterogeneity, both within and between tumors. Our study may not capture the full spectrum of GB variability, potentially limiting the applicability of our findings to all GB cases. Single time point analysis: Our study analyzed samples at a single time point, post-surgical resection. Longitudinal studies could provide more information on the dynamic changes in biomarker expression over time and during treatment. Potential confounding factors: Patient-specific factors such as age, treatment history and comorbidities could influence biomarker expression. A more detailed analysis of these factors would be beneficial. Sample size: The results could have been more significant with a bigger cohort of patients despite the gender differences found in Lumican and OPN expression in early and late middle age.

Future studies addressing these limitations could further validate and expand upon our findings, potentially leading to more robust prognostic tools for GB patients.

## 4. Materials and Methods

### 4.1. Patients and Human Samples

Samples and data from patients included in this study were provided by the Biobank IMIB (National Registry of Biobanks B. 0000859) (PT20/00109), integrated in the Platform ISCIII Biobanks and Biomodels of Spain, and they were processed following standard operating procedures with the appropriate approval of the Ethics and Scientific Committees. Histological samples were obtained from 36 patients diagnosed with glioblastoma (grade IV according to the WHO guidelines) who underwent surgical intervention at the Hospital Clinico Universitario Virgen de la Arrixaca (Murcia, Spain). Please refer to Appendix A for detailed information about the patients. Molecular markers were determined from primary tumor tissue from only untreated patients. For this study, all samples were anonymized, and no additional authorizations were required. In addition, association studies have been developed on the basis of transcriptomic data from TCGA and Rembrandt cohorts using the GlioVis database (http://gliovis.bioinfo.cnio.es/; accessed on 20 December 2024) [63].

### 4.2. Immunohistochemistry

After surgical resection, tissue specimens were fixed in 4% buffered formaldehyde (Panreac Quimica, Barcelona, Spain), embedded in paraffin, and processed by the Pathology facility (IMIB). Three-micrometer-thick serial sections were obtained from paraffin-embedded samples using an automatic rotary microtome (Thermo Scientific, Waltham, MA, USA). Sections were then incubated overnight at 4 °C with specific primary antibodies, including mouse anti-alpha-smooth muscle actin (αSMA; Ready-to-Use, IR61161-2, Dako Agilent, Santa Clara, CA, USA), rabbit anti-LAMP-2A (1/1000; 51-2200, Invitrogen, Waltham, MA, USA), goat anti-Iba-1 (1/1000; ab5076, Abcam, Amsterdam, The Netherlands), rat anti-CD68 (1/500; MCA1957T, Biorad, Hercules, CA, USA), rabbit anti-Lumican (1/200; bs-5890R, Bioss, Woburn, MA, USA) and rabbit anti-Osteopontin (1/100; 22952-1-AP, Proteintech, Manchester, UK), for staining purposes. After that, they were incubated with their respective secondary HRP or alkaline phosphatase (ALP)-labeled polymer system (Vector ImmPress, Vector Laboratories, Newark, CA, USA). The immunoreaction was revealed by using a 3,3′-diaminobenzidine (DAB) or ALP substrate kit (Dako DAB substrate kit and Vector Red Alkaline Phosphatase kit, respectively; Vector Laboratories) that identifies positive immunoreaction as a dark brown (DAB) or light red (ALP) precipitate. Finally, sections were counterstained with Mayer’s hematoxylin (LLG06272066, Carlo Erba Reagents, Barcelona, Spain). The stained markers were visualized using the Slide Viewer software (3DHISTECH) version 2.6.0.

### 4.3. Immunofluorescence and Microscopy

Three-micrometer sections of brain samples from previous studies in a GB mouse model [18] were labeled with rabbit anti-Lumican (1/200; bs-5890R, Bioss, Woburn, MA, USA) and rabbit anti-Osteopontin (1/100; 22952-1-AP, Proteintech, Manchester, UK). For fluorescent double-labeling of patient’s samples, goat anti-platelet-derived growth factor receptor beta (PDGFRβ; 1/250; BAF1042, R&D Systems, Barcelona, Spain), rabbit anti-LAMP-2A (1/1000; 51-2200, Invitrogen, Waltham, MA, USA) and mouse anti-RGS5 (1/150; MA5-25584, Invitrogen, Waltham, MA, USA) antibodies were used. Labeling was visualized by fluorescence microscopy using the corresponding secondary antibody conjugated to AlexaFluor488 (Invitrogen, Waltham, MA, USA) or Cy5 (Invitrogen, Waltham, MA, USA). Fluorescence samples were counterstained with DAPI (Invitrogen, Waltham, MA, USA) prior to mounting with Dako Fluorescence Mounting Medium (S302380-2, Dako Agilent, Santa Clara, CA, USA). A TCS-SP8-MP-AOBS laser scanning spectral inverted confocal microscope (Leica Microsystems, Wetzlar, Germany) was used to analyze the histological sections. Maximum-intensity projection of images was achieved with LAS X software version 5.2.2 (Leica Microsystems) and ImageJ software version 1.54 (NIH, New York, NY, USA).

### 4.4. Image Analysis

Images were captured from the peritumoral area, with a focus on blood vessels for perivascular analysis. All the samples studied come from a peritumoral region of the frontal lobe of the brain, including the infiltration of tumor cells that progress from the intratumoral area (core). The core and the invasion front are first determined by cellularity with hematoxylin-eosin (HE) staining, followed by tumor cell staining that is identified by the high expression of glial fibrillary acidic protein (GFAP) marker. These staining steps facilitated histology and tumor infiltration assessment. For detailed histology evaluation, please see Appendix A. Further analysis involved categorizing patients based on PC abundance with activated CMA related to the PC markers αSMA or PDGFRβ, co-localizing with the puncta pattern of the CMA lysosomal receptor LAMP-2A. Quantitative analysis of the peritumoral region was performed using ImageJ and QuPath (Version 0.5.1) software. The quantification of peritumoral areas was performed by field relative to total parenchyma. The perivascular quantification of marker CD68 was performed by encircling the blood vessels and counting the surrounding cells. Iba-1, Lumican and OPN expression was quantified by measuring DAB-positive particles per field relative to total parenchymal area and DAB-positive pixels relative to the perivascular perimeter.

### 4.5. Statistical Analysis

Statistical analysis was performed using IBM SPSS version 26. The graphs have been plotted using Graphpad Prism software (version 8.3.0). Descriptive statistics were calculated for all parameters, and patient severity was categorized into three groups—mild, intermediate, and severe—using the Mann–Whitney U-test for comparing medians. The statistical significance for multiple comparisons on continuous data was performed using a one-way ANOVA with a Scheffe post hoc test after checking that the data fitted to a normal distribution (assessed by Shapiro–Wilk or Kolmogorov–Smirnov normality test) and the difference of the variances was equal (determined by Levene’s test). In the case where our data showed unequal variances, we used T3 of the Dunnett post hoc test. Pearson’s correlation coefficient was used for assessing variable relationships. The chi-square test was used to assess the association between severity and gender. Statistical significance was set at *p* < 0.05 for two-sided tests.

## Figures and Tables

**Figure 1 ijms-26-00192-f001:**
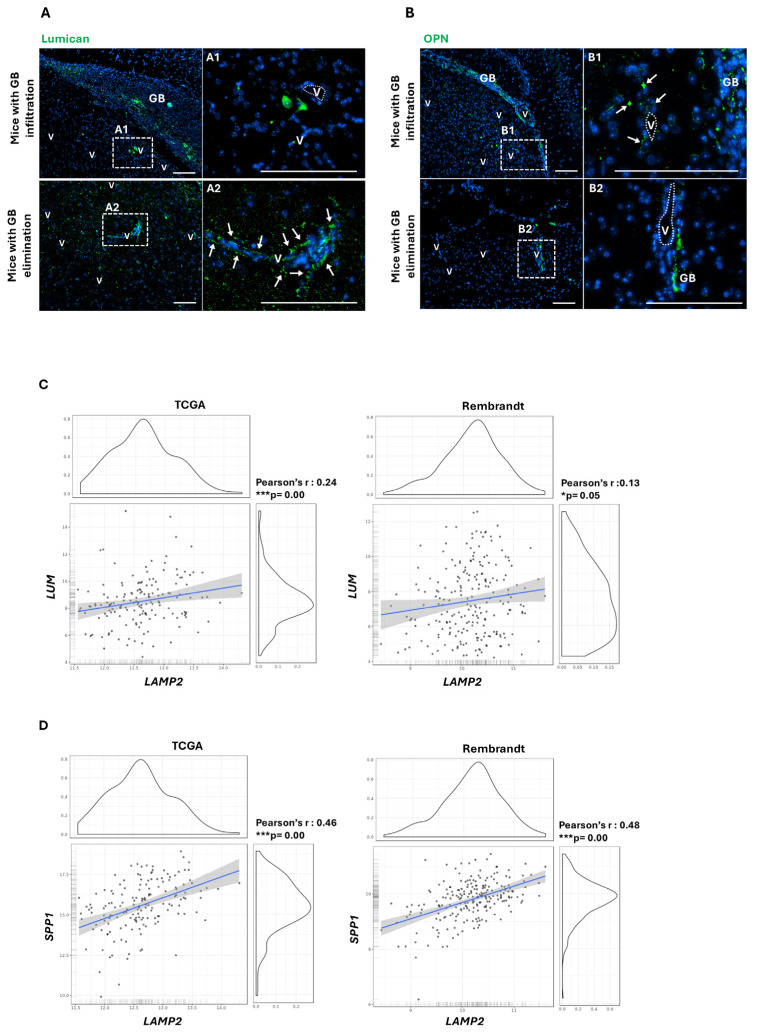
Expression of ECM proteins Lumican (**A**) and OPN (**B**) in GB xenografts from mice presenting GB infiltration or elimination. Right panels (A1 to A2 and B1 to B2) show magnification of microvessels (v) with perivascular positive stain (pointed with arrows). Scale bar: 100 µm. Association study of *LAMP2* with *LUM* (**C**) or *SPP1* (**D**) genes in the TCGA-GBM and Rembrandt cohort (GlioVis, https://gliovis.bioinfo.cnio.es/ (accessed on 20 December 2024)); * *p* < 0.05, *** *p* < 0.001.

**Figure 2 ijms-26-00192-f002:**
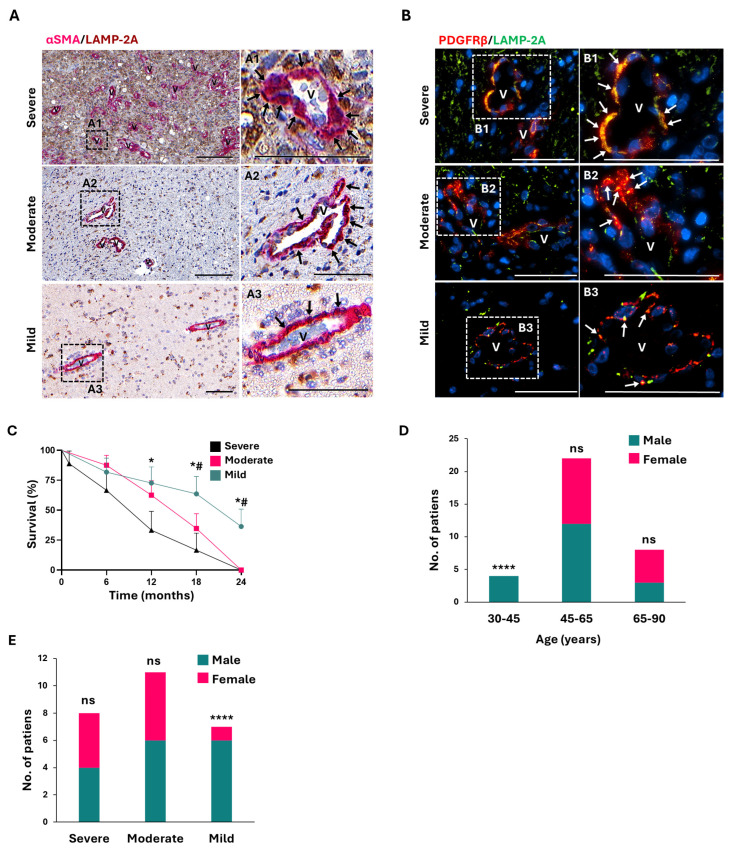
CMA activity in PCs correlates with patient survival. (**A**) Representative images of peritumoral areas according to patient classification showing co-localization of puncta pattern expression of LAMP-2A protein (dark brown) with the PC marker α-SMA (pink) in microvessels of GB patients. Samples were classified as severe (highest α-SMA/LAMP-2A co-localization), moderate or mild (basal co-localization) according to the histological evaluation. A1 to A3 shows magnifications of microvessels (v). LAMP-2A co-localizationwith α-SMA^+^ cells is marked with arrows. Scale bar: 100 µm. (**B**) Representative images of PCs marked with PDGFRβ (in red) in microvessels (v) showing co-localization of puncta pattern expression of LAMP-2A protein (in green) in peritumoral areas of severity classified GB patients. B1 to B3 show magnifications of PDGFRβ^+^ cells. Positive co-localization (in yellow) is marked with arrows. Nuclei were stained with DAPI (blue). Scale bar: 100 µm. (**C**) Overall survival shown by Kaplan–Meier curves of the severity classification related to CMA activity in PCs; * *p* < 0.05: difference between mild and severe; # *p* < 0.05: difference between mild and moderate. (**D**) Age distribution by gender of the cohort of GB patients; **** *p* < 0.0001; ns indicates no significance. (**E**) Severity classification related to gender in the age range 30–65 years in the cohort of GB patients; **** *p* < 0.0001; ns indicates no significance.

**Figure 3 ijms-26-00192-f003:**
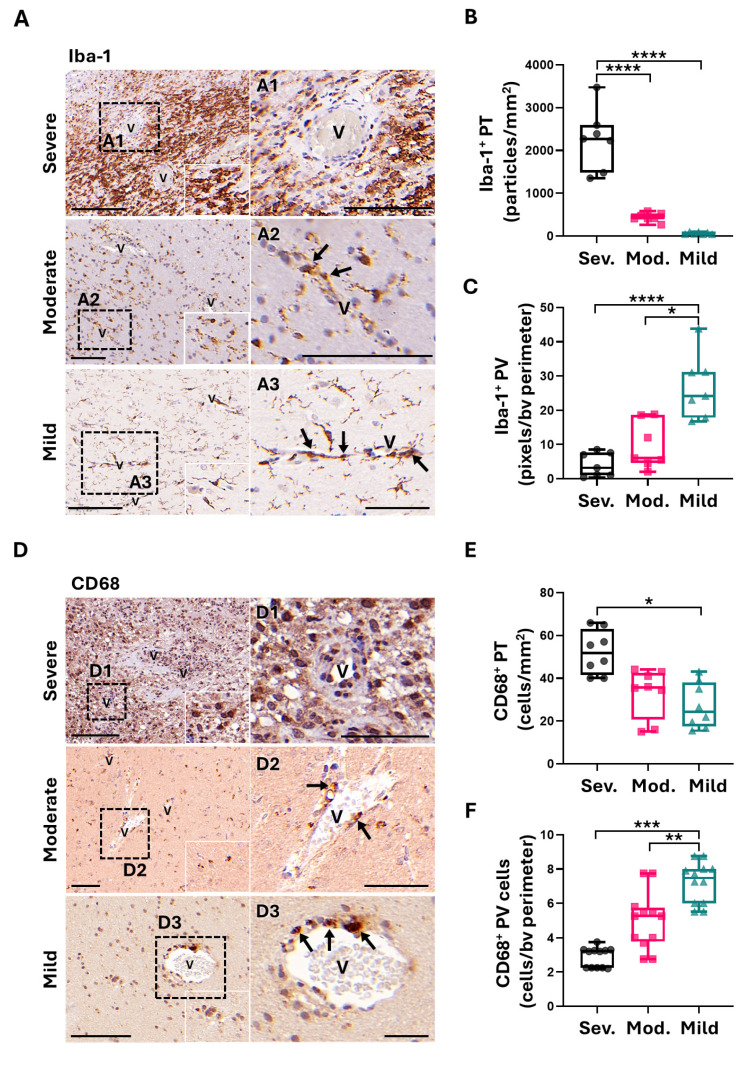
Perivascular CD68 and Iba-1 expression correlate with low CMA in peritumoral PCs and better patient outcome. (**A**) Expression of the myeloid marker Iba-1 in the peritumoral and perivascular areas (positive perivascular cells pointed with arrows) of the invasion front of GB patients. Images are representative of severe, moderate and mild grades of histopathological severity related to CMA activity in peritumoral PCs. The right corners of the left panels show an amplification of parenchymal positive cells. A1 to A3 panels show the magnification of microvessels (v) with perivascular positive cells (pointed with arrows). Scale bar: 100 µm. Boxplot diagrams of the quantification of Iba-1-positive particles related to the number of cells in the peritumoral (PT) parenchyma (**B**), expressed as positive particles per mm^2^, and in the perivascular (PV) microenvironment (**C**), expressed as positive pixels per blood vessel (bv) perimeter. Quantification was performed in at least four fields and in a minimum of 5 blood vessels; * *p* < 0.05, **** *p* < 0.0001. (**D**) Expression of phagocytic activation marker CD68 in the peritumoral and perivascular areas (positive perivascular cells pointed with arrows) of the invasion front of GB patients. Images are representative of patients classified as severe, moderate and mild related to CMA activity in peritumoral PCs. The right corners of the left panels show an amplification of parenchymal positive cells. D1 to D3 panels show magnification of microvessels (v) with perivascular positive cells (indicated with arrows). Scale bar: 100 µm. Boxplot diagrams of the quantification of CD68-positive cells in the peritumoral (PT) area (**E**), expressed as positive cells per mm^2^, and in the perivascular (PV) microenvironment (**F**), expressed as positive cells per blood vessel (bv) perimeter. Quantification was performed in at least four fields and in a minimum of 5 blood vessels; * *p* < 0.05, ** *p* < 0.01, *** *p* < 0.001. (**G**) Peritumoral and perivascular correlation between Iba-1- and CD68-positive expression. In total peritumoral parenchyma, Iba-1 and CD68 show a positive correlation (Pearson’s coefficient = 0.5640; ** *p* = 0.0077), as well as in perivascular areas (Pearson’s coefficient = 0.7827; **** *p* < 0.0001).

**Figure 4 ijms-26-00192-f004:**
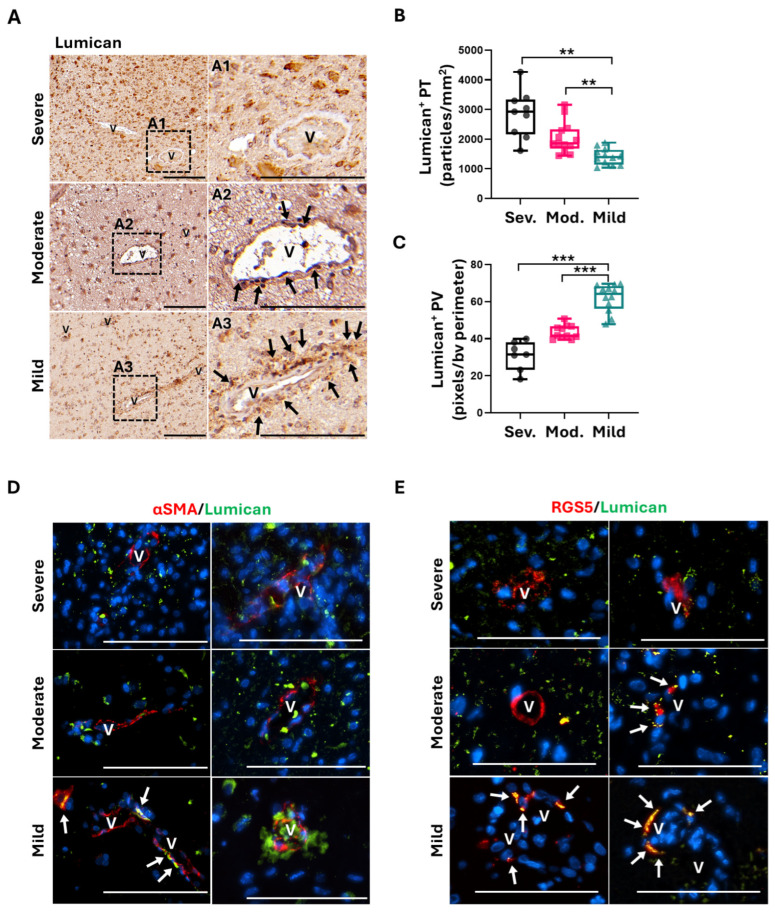
Perivascular Lumican is expressed in peritumoral areas of patients with low CMA activity in PCs. (**A**) Expression of ECM Lumican in the peritumoral and perivascular areas of the invasion front of GB. Images are representative of severe, moderate and mild grades of histopathological severity related to CMA activity in peritumoral PCs. A1 to A3 panels show magnification of microvessels (v) with perivascular positive stain (pointed with arrows). Scale bar: 100 µm. (**B**) Boxplot diagrams of the quantification of Lumican-positive particles in the peritumoral (PT) parenchyma, expressed as positive particles per mm^2^, and (**C**) in the perivascular (PV) microenvironment, expressed as positive pixels per blood vessel (bv) perimeter. Quantification was performed in at least four fields and in a minimum of 5 blood vessels; ** *p* < 0.01, *** *p* < 0.001. (**D**,**E**) Representative images of the expression of Lumican (in green) surrounding PCs (marked with αSMA or RGS5 in red) in microvessels (v) in peritumoral areas of GB patients classified by severity. PC co-localization is marked with arrows. Nuclei were stained with DAPI (blue). Scale bar: 100 µm.

**Figure 5 ijms-26-00192-f005:**
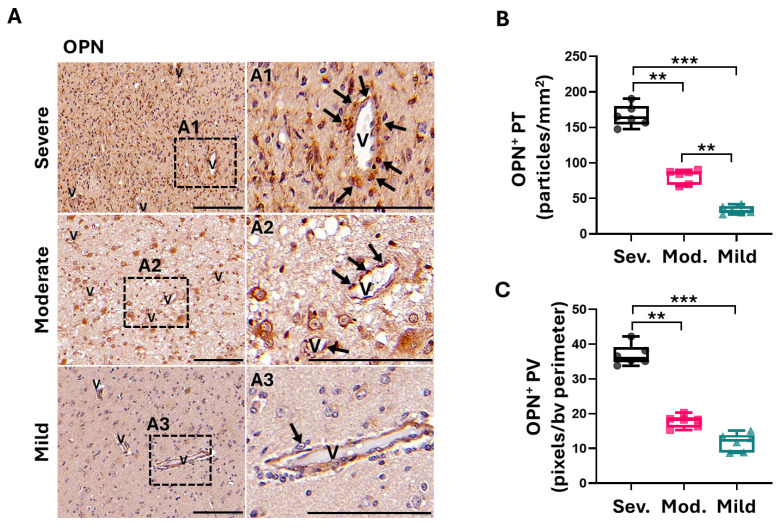
Perivascular OPN is expressed in peritumoral areas of patients with GB-induced CMA activity. (**A**) Expression of extracellular protein OPN in the peritumoral and perivascular areas of the invasion front of GB. Images are representative of patients classified as severe, moderate and mild related to CMA activity in peritumoral PCs. A1 to A3 panels show magnification of microvessels (v) with perivascular positive cells (pointed with arrows). Scale bar: 100 µm. (**B**) Boxplot diagrams of the quantification of OPN-positive cells in the peritumoral (PT) parenchyma, expressed as positive cells per mm^2^, and (**C**) in the perivascular (PV) microenvironment, expressed as positive area per blood vessel (bv) perimeter. Quantification was performed in at least four fields and in a minimum of 5 blood vessels; ** *p* < 0.01, *** *p* < 0.001. (**D**) In total peritumoral parenchyma, Lumican and OPN show a positive correlation (Pearson’s r = 0.5797; *** *p* = 0.0003), whereas in perivascular areas, they correlate negatively (Pearson’s r = −0.5098; ** *p* = 0.0047). (**E**,**F**) Representative images of the expression of OPN (in green) surrounding PCs (marked with αSMA or RGS5 in red) in microvessels (v) in peritumoral areas of GB patients classified by severity. Co-localization with PCs is indicated with arrows. Nuclei were stained with DAPI (blue). Scale bar: 100 µm.

**Table 1 ijms-26-00192-t001:** Patient cohort characteristics.

Characteristics	Value/Number (%)
Patient number	34
Gender	
Male	19 (56%)
Female	15 (44%)
Age at diagnosis, median (range)	57.33 (32.7–87.3)
Male	54.7 (32.7–79.3)
Female	60.7 (47.7–87.3)
WHO grade	Grade IV: 34 (100%)
Molecular markers	
IDH1 wt/mutation	wt (100%)
ATRX expression	28 (82%) retained
P53 expression	22 (65%) positive
CMA classification	
Severe	11 (32%)
Moderate	14 (41%)
Mild	9 (26%)

## Data Availability

The datasets used and/or analyzed during the current study are available from the corresponding author on reasonable request.

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
