# Peer review of "Expression of Lumican and Osteopontin in Perivascular Areas of the Glioblastoma Peritumoral Niche and Its Value for Prognosis"

_ijms, 2024, doi:10.3390/ijms26010192_

Round 1
Reviewer 1 Report
Comments and Suggestions for Authors
The manuscript written by Salinas et al is interesting and well written. However, major revision is required for publication in IJMS.
Comments:
- Did the authors perform a comparison between male and female patients? Did the authors find some differences between male and female patients in αSMA, LAMP-2A, Iba-1, CD68, Lumican and Osteopontin levels? If any, please describe it.
- I suggest highlighting Lumican and Osteopontin roles in cancer, especially in the context of brain cancer (If any), in the Introduction section.
- Please check carefully the text to avoid typos.
- Please check carefully all the abbreviations.
- Please add all the information related to commercial products.
Author Response
We thank the reviewer 1 for these thoughtful and valuable comments and suggestions. We have addressed them as follows:
- Did the authors perform a comparison between male and female patients? Did the authors find some differences between male and female patients in αSMA, LAMP-2A, Iba-1, CD68, Lumican and Osteopontin levels? If any, please describe it.
We have followed the reviewer’s recommendation, and we analyzed differences in incidence and severity classification (i.e. depending on GB infiltration and PC CMA related to high LAMP-2A expression levels in PCs, immune PC function related to perivascular expression of CD68 and Iba-1) in the present patient's cohort, depending on age and gender. Although our cohort is only made up of 34 patients, who we have then subdivided into three groups according to their degree of severity, we have had enough patients to find clear differences in the incidence of middle-aged men (see figure 2D). These results are supported by previous studies [1,2]. Interestingly, we have identified that in the late and early middle age range, although the incidence is higher, men seem to have a better prognosis depending on the degree of severity, while women are mostly found in the intermediate-severe groups (Figure 2E). These results, therefore, seem to indicate that women would have a worse prognosis, which is also corroborated significantly when perivascular Osteopontin biomarker is analyzed (Figure S2). Although the cohort of patients we have studied is too limited to see significant differences, the data in female patients seem to be consistent when analyzing Lumican as a marker for a good prognosis, since we practically did not find it at the perivascular level. We have described and discussed these results in the improved version of the manuscript. Please see the section of results, discussion and limitations for the study in pages 5, 9,11, 13, 14, 15:
¨Differences in incidence and survival depending on age and gender in GB have been reported [39], then we analyzed these variables along with severity classification in the present cohort. Most patients in middle age were men, as previously described (Fig. 2D), and interestingly, all except one of the female patients were included in the intermediate and severe groups. In contrast, the mild group was mostly formed by male patients (Fig. 2E)¨.
¨Analysis of perivascular Lumican revealed an interesting pattern (Fig. 4A): mild patients showed twice the levels of Lumican expression in perivascular areas than moderate and severe patients (Fig. 4C). Consistently, mild-to-moderate male patients exhibited higher perivascular levels of Lumican than females (Fig. S2A)¨.
¨Comparing between genders, the mild-to-moderate severity groups of female patients showing poor outcome at middle age (Fig. 2E), showed higher perivascular levels of OPN than males (Fig. S2B). Thus, these results also validate peritumoral OPN as an indicator of GB progression with poor prognosis ¨.
¨Furthermore, our results are supported by previous studies [39,45], showing also higher prevalence in male patients. We have found gender differences in the patient cohort when classified by severity grade (i.e. depending on PC CMA and their consequent PC immune function), specifically from early to late middle age (Fig. 2D, E). Interestingly, we found that the female patients present worse prognosis, as the cases detected were mainly classified as moderate-severe".
¨Consistent with previous results on the female patient tendency to present worse outcomes (Fig. 2E), in particular in middle age groups, we found that female patients exhibit lower levels of Lumican and higher levels of OPN at the perivascular level (Fig. S2). Consequently, these results might add molecular background to the finding of the poorer prognosis of female patients¨.
¨Sample size: the results could have been more significant with a bigger cohort of patients despite the gender differences found in Lumican and OPN expression, in early and late middle age¨.
- I suggest highlighting Lumican and Osteopontin roles in cancer, especially in the context of brain cancer (If any), in the Introduction section.
We completely agree with the reviewer’s suggestion. Please, note the changes introduced in the introduction (see page 2) and discussion (see page 14) sections following the reviewer’s recommendation.
¨In the context of brain cancers, the relevance of Lumican remains poorly studied. The increased expression of intracellular Lumican in GB and neuroblastoma cells has been linked to the preservation of a stem-like phenotype, drug (temozolomide)-resistant, and reduced overall survival [34,35].By opposite, in children medulloblastoma, extracellular Lumican is detected in the low-risk group but not in aggressive tumors [36].
On the other hand, OPN is a multifunctional secreted phosphorylated glycoprotein with adhesion sequences to interact with ECM components and cell surface integrins. By these interactions, OPN regulates chemotactic migration and survival of macrophages [20–22]. In the context of brain tumors, and in GB in particular, OPN secreted by tumor cells and GAMMs, contributes to the recruitment of macrophages to the GAMMs pool and to the M2 polarization maintenance, which in turn compromises the anti-tumor immune responses, favors angiogenesis, and facilitates tumor evasion [21,22,37] ¨.
¨A recent study in medulloblastoma has described that Lumican can be specifically located on the periphery of nodules of the low-risk subset of medulloblastoma, but it is absent on the nodules of the most aggressive subtype, frequently associated with metastases [36]. Related to this study, anti-tumoral PCs in response to GB cells secrete elevated levels of Lumican, as well as other proteins with anti-tumoral properties, which might contribute to reduce GB progression [18]¨.
- Please check carefully the text to avoid typos.
We apologize for this mistake. Please, note that we have corrected different typos along the new text.
- Please check carefully all the abbreviations.
We apologize for this mistake. Please, note that we have corrected it in the improved version of the manuscript.
- Please add all the information related to commercial products.
We have followed the reviewer’s recommendation, and we have included all the information related to commercial products. Please note the highlighted text along the methods section.

Reviewer 2 Report
Comments and Suggestions for Authors
The current manuscript entitled “Expression of Lumican and Osteopontin in Perivascular Areas of the Glioblastoma Peritumoral Niche and its Value for Prognosis” by Salinas et al is focused on exploring the tumor microenvironment in gliomas. Authors have explored the correlation between chaperone-mediated autophagy (CMA) activity in pericytes (PCs) promoting immune suppression and glioma infiltration. Authors have also examined the role of extra cellular matrix (ECM) and secreted proteins and their impact in glioma progression. Moreover, authors have proposed Lumican and Osteopontin as a prognostic marker to identify perivascular areas of gliomas. The present study seems extension the previous report from the same group (DOI: 10.3389/fcell.2022.797945). I have few specific suggestions:
Functional validation required in mouse model at least (since authors have model to provide strong evidence for this hypothesis, Ref- 18)
Detailed information about Expression profile (WES/ RNASeq) would be important for these studies
Did authors have information of tumor location (core vs invasion) to decide the number of pericytes and difference between immune compartment between edge and core
Authors should provide explanation to determine CMA classification (Ref)
Kaplan-Meier curves require proper annotations (not averages survival percentage) (points on Y- axis?) and statistical analysis is missing.
I would advise authors to provide some validation experiments with interference in Lumican and Osteopontin in glioma cells progression
Authors can also explore the public data set in accordance with their hypothesis
Author Response
We thank the reviewer 2 for these thoughtful and valuable comments and suggestions. We have addressed them as follows:
- Functional validation required in mouse model at least (since authors have model to provide strong evidence for this hypothesis, Ref- 18).
We thank the reviewer’s request. As suggested, we have performed immunohistochemical analysis in the mice brain samples from the study of Molina et al. 2022 Front Dev Cell Biol. Please see the new figure 1 and results section in page 3 of the improved manuscript.
¨2.1 Validation of Lumican and OPN as GB prognosis biomarkers in a mouse model with GB infiltration dependent on GB-induced CMA in PCs.
We previously had shown in a mass spec study that OPN is included in the GB-conditioned PC secretome, which is dependent on aberrantly induced CMA in PCs [8] and is composed mainly by proteins implicated in the pro-tumor immune responses [18]. By opposite, Lumican secretion was found to be enriched in the secretome of CMA deficient PCs responding to GB cells, among other molecules contributing to the anti-tumor immune responses [18]. Moreover, our findings revealed that CMA deficient PCs through their secretome can eliminate GB tumor growth, whereas competent CMA PCs promote GB progression [8,18]. Concluding with these studies, we determined that some molecules in the PC secretome, along with the expression of genes related to their phenotype, might be used as possible prognostic markers [18]. Thus, we decided to validate Lumican and OPN in the perivascular areas, including PCs, as GB progression prognostic markers. Firstly, we tested their expression in the GB peritumoral niche of our GB mouse model presenting GB infiltration, or GB elimination, depending on competent or deficient CMA PCs respectively.
As expected, brain areas with GB tumor mass only showed Lumican expression within the tumor cell mass and in the peritumoral GB cell infiltration. In contrast, Lumican was significantly identified in perivascular areas surrounding brain parenchyma where there was a previous tumor mass and infiltrated GB elimination (Fig. 1A). Conversely, OPN was found in peritumoral microvessels of GB mice, supporting tumor cell growth and survival through GB-induced CMA in PCs. By opposite, mice presenting GB elimination just showed OPN expression in some remanent tumor cells (Fig. 1B). With this first approach we had proven that Lumican and OPN could be biomarkers of GB progression dependent on GB-induced CMA in PCs¨.
- Detailed information about Expression profile (WES/ RNASeq) would be important for these studies.
This study is based on a previous comparative RNAseq and proteomics studies showing a gene expression profile dependent on GB-induced CMA in conditioned PCs by tumor cells compared to an anti-tumoral PC phenotype without CMA. In this work we confirmed protein expression and function of some of these genes [3]. Thus, we believe it is not necessary to describe the expression profile (WES/RNAseq) in this work, as it clearly continues the previous one which appears referenced throughout the entire text. In any case, we have described this work in more detail at the beginning of the results section (page 3), in line with the reviewer's suggestion.
¨Additionally, the RNAseq study of the GB-conditioned PCs compared to CMA deficient PCs in response to GB cells, revealed gene expression profiles which supported previous results on the protumoral anti-inflammatory and anti-tumoral inflammatory phenotypes of PCs, respectively. Moreover, our findings revealed that CMA deficient PCs through their secretome can eliminate GB tumor growth, whereas competent CMA PCs promote GB progression [8,18]. Concluding with these studies, we determined that some molecules in the PC secretome, along with the expression of genes related to their phenotype, might be used as possible prognostic markers [18]¨.
- Did authors have information of tumor location (core vs invasion) to decide the number of pericytes and difference between immune compartment between edge and core
We thank the reviewer´s comment. We have accordingly improved the text in the methods and the discussion sections. Please see pages 13 and 16:
“All the samples studied come from a peritumoral region of the frontal lobe of the brain including infiltration of tumor cells that progress from the intratumoral area (core). The core and the invasion front are determined firstly by cellularity with Hematoxylin-eosin (HE) staining, followed by tumor cell staining that is identified by high expression of glial fibrillary acidic protein (GFAP) marker. These staining steps facilitated histology and tumor infiltration assessment.”
¨We do not know if the PCs of the tumor core and the peritumoral zone are different in number, but they might as we have previously shown [6,8] that GB-conditioned PCs acquire immunosuppressive properties and their proliferation is affected. We believe that PCs in the tumor core or invasion front seem equal, but their immune functions can present differences depending on grade of GB invasion. Then, the immune compartment is determined by the amount of GB cells conditioning PCs though aberrantly induced CMA. In the core, all PCs are conditioned by complete GB invasion of the microvasculature, whereas in the microvasculature of peritumoral regions where GB cells infiltrate, there are still non-conditioned PCs responding to the tumor. Thus, immune properties of PCs, and consequently the surrounding tumor cell microenvironment could change between both edge and core compartments [8,45].¨
- Authors should provide explanation to determine CMA classification (Ref)
We have improved the manuscript text according to the reviewer’s request. Please, see the results section in page 5:
¨As we have previously shown that GB-induced CMA in PCs is required for tumor cell survival and tumor progression [8,18], we checked the patient severity classification according to GB-induced CMA in PCs located in peritumoral microvessels, where GB cells infiltrate from the invasion front (PNAS). Then, we analyzed the expression levels of the CMA lysosomal receptor, LAMP-2A, in peritumoral PCs (PC markers; α-SMA or PDGFRβ) (Fig. 2 and S1).
Severe patients displayed hypercellularity with elevated astroglia-invaded parenchyma that correlated with the appearance of aberrant blood vessels (Fig. S1). In the peritumoral areas of these patients, PCs showed upregulated levels of CMA activity (Fig. 2A and B), as has been previously shown [8]. Similarly, moderate classified patients exhibited large parenchyma invasion with GB cells becoming round in the peritumoral areas (Fig. S1B). In these patients, CMA activity was still high in PCs (Fig. 2A and B), but the perivascular areas were not yet fully invaded by tumor cells (Fig S1B). In contrast, mild patients displayed a regular brain parenchyma with few invaded blood vessels (Fig. 2A, S1), which was related to an expected normal LAMP-2A expression in PCs (Fig. 2A and B). Consequently, the severity grade of patients, and therefore, the GB-induced CMA activity of their peritumoral PCs, was correlated with the cohort survival (Fig. 2C)¨.
- Kaplan-Meier curves require proper annotations (not averages survival percentage) (points on Y- axis?) and statistical analysis is missing.
We apologize for this mistake. Please, see the updated figure 2.
6. I would advise authors to provide some validation experiments with interference in Lumican and Osteopontin in glioma cells progression
We thank the reviewer for the valuable comment. We had already seen in a mouse model that the progression of glioblastoma is dependent on the PC CMA, and consequently on its immune function. With this study we just pretend to validate that Lumican and Osteopontin, among other markers such as LAMP-2A, Iba and CD68, are potential biomarkers at the perivascular level of the peritumoral regions surrounding the glioblastoma invasion front. We completely agree with the fact that a functional study to demonstrate the specificity of these proteins would have a high impact and might validate their roles affecting glioma progression as targets for GB therapy. However, we believe a functional study of the expression of these proteins would be also complicated as other cells from the tumor microenvironment such as microglia might have indirect effects affected by PC immune function and could be also implicated in the secretion of these proteins. Additionally, the roles of both proteins might be different when tumor cells expressed them or microenvironment cells secrete them with immune properties, and in this case interference of proteins should be specifically in pericytes as we are just focusing on perivascular expression related to the secreted protein by PC-GB dependent on CMA regulation in PCs. Importantly, secreted proteins by microenvironment cells have been reported to have different effects on tumor cells than those proteins expressed by tumor cells themselves [4–8]. This study would be very valuable to go deeper into other different studies to analyze Lumican and OPN, as possible targets for glioblastoma progression, but we think that this is not the study goal of this work.
- Authors can also explore the public data set in accordance with their hypothesis
We completely agree with the reviewer’s suggestion. Following the reviewer’s recommendation, we have performed an analysis of Lumican and OPN gene expression correlated with Lamp2 gene expression in the TCGA and Rembrandt cohorts through the GlioVis platform. Please, see figure 1C, 1D and updated results and methods section in page 3 and 15.
¨To support our results, we aimed to further characterize the link between CMA and Lumican and OPN expression in clinical samples from public datasets (TCGA and Rembrandt cohorts). These datasets had previously shown to present high levels of LAMP2A gene expression, which positively correlated with metabolism, proliferation, and ECM interaction markers in glioblastoma stem cells (GSCs) [38].
Importantly, data of GB samples from these two cohorts revealed a positive correlation between LAMP2 gene expression and both Lumican (LUM) (Fig. 1C) and OPN (SPP1) genes (Fig. 1D). Our findings conclusively validated both perivascular Lumican and OPN expression, probably dependent on PC CMA, as biomarkers of prognosis of GB progression¨.
¨In addition, association studies have been developed on the basis of transcriptomic data from TCGA and Rembrandt cohorts using GlioVis database (http://gliovis.bioinfo.cnio.es/) [63]¨.
References
- Tian, M.; Ma, W.; Chen, Y.; Yu, Y.; Zhu, D.; Shi, J.; Zhang, Y. Impact of Gender on the Survival of Patients with Glioblastoma. Bioscience Reports 2018, 38, BSR20180752, doi:10.1042/BSR20180752.
- Lee, J.; Nicosia, M.; Hong, E.S.; Silver, D.J.; Li, C.; Bayik, D.; Watson, D.C.; Lauko, A.; Kay, K.E.; Wang, S.Z.; et al. Sex-Biased T-Cell Exhaustion Drives Differential Immune Responses in Glioblastoma. Cancer Discovery 2023, 13, 2090–2105, doi:10.1158/2159-8290.CD-22-0869.
- Molina, M.L.; García-Bernal, D.; Salinas, M.D.; Rubio, G.; Aparicio, P.; Moraleda, J.M.; Martínez, S.; Valdor, R. Chaperone-Mediated Autophagy Ablation in Pericytes Reveals New Glioblastoma Prognostic Markers and Efficient Treatment Against Tumor Progression. Frontiers in Cell and Developmental Biology 2022, 10, 797945, doi:https://doi.org/10.3389/fcell.2022.797945.
- Li, X.; Truty, M.A.; Kang, Y.; Chopin-Laly, X.; Zhang, R.; Roife, D.; Chatterjee, D.; Lin, E.; Thomas, R.M.; Wang, H.; et al. Extracellular Lumican Inhibits Pancreatic Cancer Cell Growth and Is Associated with Prolonged Survival after Surgery. Clinical Cancer Research 2014, 20, 6529–6540, doi:10.1158/1078-0432.CCR-14-0970.
- Appunni, S.; Anand, V.; Khandelwal, M.; Gupta, N.; Rubens, M.; Sharma, A. Small Leucine Rich Proteoglycans (Decorin, Biglycan and Lumican) in Cancer. Clin Chim Acta 2019, 491, 1–7, doi:10.1016/j.cca.2019.01.003.
- Wei, J.; Marisetty, A.; Schrand, B.; Gabrusiewicz, K.; Hashimoto, Y.; Ott, M.; Grami, Z.; Kong, L.-Y.; Ling, X.; Caruso, H.; et al. Osteopontin Mediates Glioblastoma-Associated Macrophage Infiltration and Is a Potential Therapeutic Target. J Clin Invest 2019, 129, 137–149, doi:10.1172/JCI121266.
- Kariya, Y.; Kariya, Y. Osteopontin in Cancer: Mechanisms and Therapeutic Targets. International Journal of Translational Medicine 2022, 2, 419–447, doi:10.3390/ijtm2030033.
- Hao, C.; Lane, J.; Jiang, W.G. Osteopontin and Cancer: Insights into Its Role in Drug Resistance. Biomedicines 2023, 11, 197, doi:10.3390/biomedicines11010197.

Round 2
Reviewer 1 Report
Comments and Suggestions for Authors
The authors addressed all the reviewers requests.
Reviewer 2 Report
Comments and Suggestions for Authors
The authors have carefully integrated the feedback, which has greatly enhanced the manuscript. I acknowledge their efforts in effectively incorporation the recommendations leading to significant improvement in the overall quality of the work.